# Reconstruction Attacks on Machine Unlearning: Simple Models are Vulnerable

**Martin Bertran** *
Amazon AWS AI/ML

**Shuai Tang** *
Jump Trading

**Michael Kearns**
University of Pennsylvania
Amazon AWS AI/ML

**Jamie Morgenstern**
University of Washington
Amazon AWS AI/ML

**Aaron Roth**
University of Pennsylvania
Amazon AWS AI/ML

**Zhiwei Steven Wu**
Carnegie Mellon University
Amazon AWS AI/ML

## Abstract

*Machine unlearning* is motivated by desire for data autonomy: a person can request to have their data's influence removed from deployed models, and those models should be updated as if they were retrained without the person's data. We show that, counter-intuitively, these updates expose individuals to high-accuracy *reconstruction attacks* which allow the attacker to recover their data in its entirety, even when the original models are so simple that privacy risk might not otherwise have been a concern. We show how to mount a near-perfect attack on the deleted data point from linear regression models. We then generalize our attack to other loss functions and architectures, and empirically demonstrate the effectiveness of our attacks across a wide range of datasets (capturing both tabular and image data). Our work highlights that privacy risk is significant even for extremely simple model classes when individuals can request deletion of their data from the model.

## 1 Introduction

As model training on personal data becomes commonplace, there has been a growing literature on data protection in machine learning (ML), which includes at least two aspects:

**Data Privacy** The primary concern regarding data privacy in machine learning (ML) applications is that models might inadvertently reveal details about the individual data points used in their training. This type of privacy risk can manifest in various ways, ranging from membership inference attacks (Shokri et al., 2017)—which only seek to confirm whether a specific individual's data was used in the training—to more severe reconstruction attacks (Dick et al., 2023) that attempt to recover entire data records of numerous individuals. To address these risks, algorithms that adhere to differential privacy standards (Dwork et al., 2006) provide proven safeguards, specifically limiting the ability to infer information about individual training data.

**Machine Unlearning** Proponents of data autonomy have advocated for individuals to have the right to decide how their data is used, including the right to retroactively ask that their data and its influences be removed from any model trained on it. Data deletion, or *machine unlearning*, refer to technical approaches which allow such removal of influence (Ginart et al., 2019; Cao & Yang, 2015). The idea is that, after an individual's data is deleted, the resulting model should be in the state it would have been had the model originally been trained without the individual in question's data. The primary

---

[0]Martin and Shuai are the lead authors, and other authors are ordered alphabetically. Work done while Shuai was employed by Amazon.{maberlop AT amazon.com}

38th Conference on Neural Information Processing Systems (NeurIPS 2024).

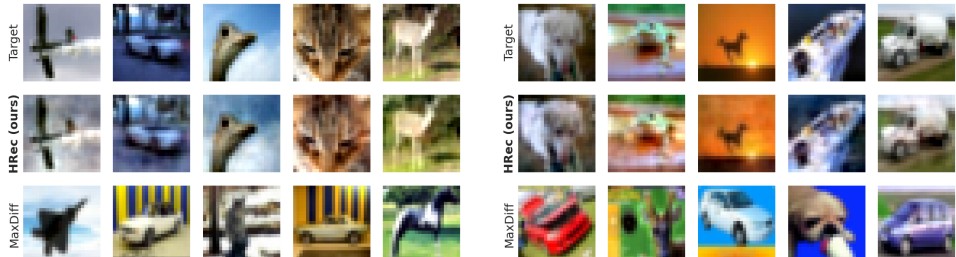

Figure 2: CIFAR10 samples reconstructed from a logistic regression model over a random Fourier feature embedding (4096) of the raw input. We randomly chose one deleted sample per label (Row 1) and compared them against the reconstructed sample using our method (HRec, Row 2) and a perturbation baseline (MaxDiff, Row 3) which searches for the public sample with the largest prediction difference before and after sample deletion. HRec produces reconstructions similar to the deleted images both visually and quantitatively measured by cosine similarity.

focus of this literature has been on achieving or approximating this condition for complex models in ways that are more computationally efficient than full retraining (see e.g. Golatkar et al. (2020); Izzo et al. (2021); Gao et al. (2022); Neel et al. (2021); Bourtoule et al. (2021); Gupta et al. (2021).)

Practical work on both privacy attacks (like membership inference and reconstruction attacks) and machine unlearning has generally focused on large, complex models like deep neural networks. This is because (1) these models are the ones that are (perceived as) most susceptible to privacy attacks, since they have the greatest capacity to memorize data, and (2) they provide the most technically challenging case for machine unlearning (since for simple models, the baseline of just retraining the model is feasible). For simple (e.g., linear) models, the common wisdom has been that the risk of privacy attacks is low, and indeed, we verify in Appendix A that state-of-the-art membership inference attacks fail to achieve non-trivial performance when attacking linear models trained on tabular data, and an example is shown in Figure 1.

The main message of our paper is that the situation changes starkly when we consider privacy risks in the presence of machine unlearning. As we show, absent additional protections like differential privacy, requesting that your data be removed—even from a linear regression model—can expose you to a complete reconstruction attack. Informally, this is because it gives the adversary *two models* that differ in whether your data was used in training, which allows them to attempt a differencing attack. We show that the parameter difference between the two models can be approximately (or exactly, for linear models) expressed as a function of the gradient of the deleted sample and the expected Hessian of the model w.r.t. public data. This allows us to *equate model unlearning to releasing the gradient of the unlearned samples*, and leverage existing literature on reconstruction from sample gradients to achieve our results.

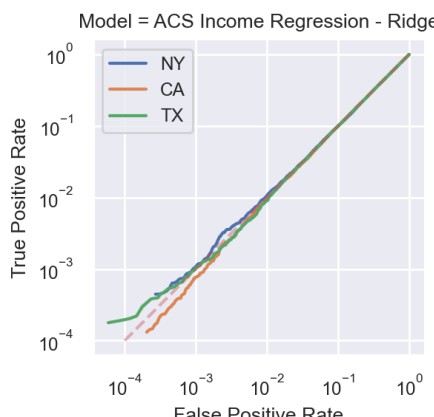

Figure 1: We conduct membership inference attacks on a ridge regression on ACS Income task; the attack performance is poor (close to random guessing).

**Our Contributions** We consider the following threat model: An attacker has access to the parameters of some model both before and after a deletion request is made. The attacker also has sampling access to the underlying data distribution, but no access to the training set of the models. In this setting, we first study exact reconstruction attacks on linear regression. We give an attack that accurately recovers the deleted sample given the pair of linear models before and after sample deletion. This is made possible by leveraging the closed-form single-sample training algorithm for linear regression as well as the ability to accurately estimate the covariance matrix of the training data from a modestly sized disjoint sample of public data drawn from the same distribution.

We then extend our attack to the setting where the model consists of a (fixed and known) embedding function, followed by a trained linear layer. Our goal remains to recover the original data point (not only its embedding). This is a natural class of simple models, and also captures last-layer fine-tuning on top of a pre-trained model - a common setting in which machine unlearning guarantees are offered.

Finally, we give a second-order unlearning approximation via Newton's method which extends our attack to generic loss functions and model architectures. This provides a way to approximate the gradient of a deleted sample with respect to the model parameters, and later the sample itself. We remark that Newton's update approximation has itself been proposed as a way to approximate unlearning (Izzo et al., 2021; Gao et al., 2022), and is naturally related to the literature on influence functions (Hampel, 1974; Koh & Liang, 2017; Zhang & Zhang, 2022), which examines the effect any (set of) samples has on the final trained model.

We experimentally demonstrate the effectiveness of our attack on a variety of simple tabular and image classification and regression tasks. The success of our attack highlights the privacy risks of retraining models to remove the influence of individual's data, without additional protections like differential privacy (as advocated by e.g. Chourasia et al. (2022) in another context). Figure 2 shows several deleted samples from a model trained on CIFAR10 (Krizhevsky et al., 2009) alongside the recovered reconstructions for our method and a baseline based on public data.

## 1.1 Additional Related Work

We elaborate on our threat model and related work to position our approach within the existing literature. Our model involves a scenario where a model maintainer trains a model on a private dataset, $X_{\text{priv}} \in \mathbb{R}^{n \times d}$ and $y_{\text{priv}} \in \mathbb{R}^n$, to minimize a loss function $\ell$, yielding parameters $\beta^+$. Upon a user's request for data deletion, the maintainer re-trains the model excluding the user's data, resulting in new parameters $\beta^-$. Our adversary, equipped with both model parameters before and after the deletion $\beta^+, \beta^-$ and access to public samples $(X_{\text{pub}}, y_{\text{pub}})$ from the same distribution, aims to reconstruct the deleted sample $(x, y)$ using the algorithm $\mathcal{A}(\beta^+, \beta^-, X_{\text{pub}}, y_{\text{pub}}) \rightarrow (\tilde{x}, \tilde{y})$.

Extensive literature exists on membership inference and reconstruction attacks against static models, with notable references including Shokri et al. (2017); Carlini et al. (2021, 2022); Bertran et al. (2023); Carlini et al. (2023). These studies primarily address attacks on large-scale models, whereas our work focuses on simpler models and explores the changes induced by deletion operations.

Pioneering work in model inversion attacks on linear models by Fredrikson et al. (2014); Wu et al. (2015) demonstrates that partial information about a data sample can be inferred from model outputs. However, these attacks do not directly utilize model parameters.

The concept of privacy risks in machine unlearning was first outlined by Chen et al. (2021), who introduced a membership inference attack based on shadow models applicable to updates in machine learning models. This attack's efficacy increases with model complexity. In contrast, our approach targets the reconstruction of the exact data point, extending beyond mere membership inference.

Further research by Salem et al. (2020) explored reconstruction attacks using single-gradient updates on complex models. Unlike their approach, which relies solely on API access to the model, our method utilizes direct access to model parameters, allowing for efficient reconstruction of data points even in fully retrained simple models.

Recent works have also examined gradient-based reconstruction attacks, typically focusing on untrained models (Zhu et al., 2019; Zhao et al., 2020; Wang et al., 2023). Our contribution extends these techniques to the context of machine unlearning, highlighting the utility of analyzing updates for data reconstruction.

Balle et al. (2022) present a scenario where the adversary possesses almost complete knowledge of the training dataset, aiming to reconstruct the missing sample. This represents a distinct model from ours, where the adversary's knowledge is limited to model parameters and public samples.

Overall, our work contributes to the understanding of privacy vulnerabilities in machine learning, particularly in scenarios involving model updates and unlearning, where adversaries exploit the slight yet informative differences between model parameters. Most work on machine unlearning asks that they "unlearned" models be indistinguishable from what would have been obtained had the model been retrained without the deleted points — the baseline that we attack in this paper. However

there are several exceptions Cohen et al. (2023); Garg et al. (2020) that ask for stronger conditions (satisfied, e.g. by requiring that the entire sequence of models released satisfy differential privacy like conditions) that can preclude such attacks. The concurrent work in Hu et al. (2024) shares thematic similarities by attacking specific (linearized) approximations to model unlearning.

## 2    Method

We develop an attack aimed at reconstructing the features of a deleted user from a (regularized) linear regression model. Specifically, our focus is on reconstructing a sample $(x, y)$, previously part of the private training dataset $X_{\text{priv}}, y_{\text{priv}}$, using models trained before and after the deletion of this sample.

The parameters of these models, $\beta^+$ and $\beta^-$, are solutions to a regularized linear regression problem. The model including the deleted sample yields:

$$\beta^+ = \arg \min_{\beta} \|X_{\text{priv}}\beta - y_{\text{priv}}\|_2^2 + \lambda\|\beta\|_2^2, \tag{1}$$

with a closed-form solution $\beta^+ = C^{-1} X_{\text{priv}}^\top y_{\text{priv}}$. Here, $C = X_{\text{priv}}^\top X_{\text{priv}} + \lambda I$ represents the regularized covariance matrix.

For the model post-deletion, $\beta^-$ can be described similarly, but adjusted for the absence of $(x, y)$:

$$\beta^- = (C - xx^\top)^{-1}(X_{\text{priv}}^\top y_{\text{priv}} - x^\top y). \tag{2}$$

Using the Sherman-Morrison formula, we can relate $\beta^+$ and $\beta^-$ via:

$$\beta^+ = \beta^- + \frac{y - x^\top \beta^-}{1 + x^\top C^{-1} x} C^{-1} x. \tag{3}$$

This equation leads to an expression for the change in parameters due to the deletion of $(x, y)$:

$$C(\beta^+ - \beta^-) = \alpha(x, y)x, \tag{4}$$

where $\alpha(x, y)$ is a scalar function dependent on the sample. This representation shows that the difference between $\beta^+$ and $\beta^-$, scaled by $C$, is proportional to $x$, which suggests a potential avenue for reconstructing $x$.

However, when we do not have access to $X_{\text{priv}}$ or $\lambda$, we must rely on publicly available data $X_{\text{pub}}, y_{\text{pub}}$. We approximate $C$ using public data as $\hat{C} = X_{\text{pub}}^\top X_{\text{pub}}$, considering that $C$ and $\hat{C}$ are both empirical estimates of the underlying statistical covariance $\mathbb{E}[x^t x]$. For a rigorous analysis on bounding the errors in this approximation, see Tropp et al. (2015).

Linear models are often learned with bias terms, which can be interpreted as a feature with value 1. To simply the notation, we assume that the $d$-th dimension of a sample $x$ is 1, noted as $x_d = 1$. Given our assumption, the scaling factor $\alpha(x, y)$ is adjusted by normalizing the reconstructed feature vector to ensure the scale of $x$ is maintained. We then estimate the reconstructed sample $(\tilde{x}, y)$ as follows:

$$\tilde{x} = \hat{z}/\hat{z}_d, \qquad \text{where} \qquad \hat{z} = \hat{C}(\beta^+ - \beta^-) \tag{5}$$

where $\hat{z}_d$ is the $d$-th element of $\hat{z}$, ensuring $\tilde{x}_d = 1$. This method offers a systematic approach to estimate deleted user features from differential changes in model parameters, employing public data to approximate necessary statistics and regularization impacts.

## 3    Beyond Linear Regression

Our attack is derived for the simple models used in linear regression. However, the same ideas can be generalised beyond linear regression. The attack generalises immediately to any model which performs linear regression on top of a fixed embedding: our attack recovers the embedding of the deleted point, and reduces the problem to inverting the embedding. We can also generalise to other loss functions—The primary challenge is that we no longer have closed-form expressions for an "update", but we can approximate this, as we describe in Section 3.2.

## 3.1 Fixed Embedding Functions

Our attack is built upon the analytical update of adding or deleting a sample to a linear regression model, therefore, it also generalises to linear models trained on top of embeddings of the original features. Suppose that both parameter vectors $\beta^+$ and $\beta^-$ along with the embedding function $\phi : \mathbb{R}^d \to \mathbb{R}^{d'}$ are publicly known (and the embedding function has a bias term). Our attack first reconstructs the embeddings as in Eq. (5), and then reconstructs features by finding a data point whose embedding best matches the reconstructed transformed features as follows:

$$\tilde{x} = \arg\min_x \|\tilde{z}/\tilde{z}_{d'} - \phi(x)\|, \qquad \text{where} \qquad \tilde{z} = \phi(X_{\text{pub}})^\top \phi(X_{\text{pub}}) \left(\beta^+ - \beta^-\right) \qquad (6)$$

Here, we assume that the embedding $\phi$ is fixed—that is, it doesn't change after deleting a sample. This is the case when e.g. performing last-layer fine-tuning on top of a pre-trained model, and for data-independent embeddings like random Fourier features.

## 3.2 Arbitrary Loss Functions and Architectures

Our foundational equation, Eq. (4), specifically addresses linear models minimized under the mean squared error. When broadening this scope to include alternative loss functions and model architectures, the luxury of closed-form solutions vanishes. However, we can utilize Newton's method for approximating the "update function" necessary after data deletion. Consider a model maintainer optimizing an empirical risk function represented as:

$$\ell(\beta; X_{\text{priv}}, y_{\text{priv}}) = \frac{1}{n} \Sigma_{(x,y) \in \{X_{\text{priv}}, y_{\text{priv}}\}} \ell(\beta; x, y), \qquad (7)$$

where $\beta^+$ and $\beta^-$ are the optimal parameters before and after excluding a specific data point, $(x, y)$, respectively. By adopting a second-order Taylor approximation via Newton's method, we estimate:

$$\beta^- \approx \beta^+ - H^{-1} \nabla \ell, \qquad (8)$$
$$\text{where } H = \nabla^2_{\beta=\beta^+} \ell(\beta; X_{\text{priv}} \backslash x, y_{\text{priv}} \backslash y),$$
$$\nabla \ell = \nabla_{\beta=\beta^+} \ell(\beta; X_{\text{priv}} \backslash x, y_{\text{priv}} \backslash y).$$

Using the first-order optimality conditions, we deduce that the aggregate gradient over the remaining samples inversely equals that of the removed sample:

$$\nabla_{\beta=\beta^+} \ell(\beta; X_{\text{priv}}, y_{\text{priv}}) = 0, \qquad (9)$$
$$n\nabla\ell = -\nabla_{\beta=\beta^+} \ell(\beta; x, y). \qquad (10)$$

Integrating Eq. (10) into Eq. (8), we derive:

$$\beta^- \approx \beta^+ + \frac{H^{-1}}{n} \nabla_{\beta=\beta^+} \ell(\beta; x, y), \qquad (11)$$
$$nH(\beta^+ - \beta^-) \approx -\nabla_{\beta=\beta^+} \ell(\beta; x, y). \qquad (12)$$

In linear regression, this method precisely recovers the known Eq (4). In the general case, the Hessian matrix, analogous to the covariance matrix in linear regression, is estimated using public data sharing the same distribution as the private data:

$$\hat{H} = \frac{1}{m} \Sigma_{x', y' \in \{X_{\text{pub}}, y_{\text{pub}}\}} \nabla^2_{\beta=\beta^+} \ell(\beta; x', y'). \qquad (13)$$

For linear models under attack, the gradients at the removed loss sample, particularly for certain model layers, correlate directly with the data of the removed sample. Efficiently approximated through influence matrices such as the Fisher information matrix, these gradients serve as crucial elements in reconstructive attacks on model privacy.

For non-linear models and more intricate architectures, the materialization of the Hessian may become impractical due to its size, prompting the use of efficient Hessian-vector product computations.

---

**Algorithm 1** Generalized Attack

---

**Require:** Public data $X_{\text{pub}} \in \mathbb{R}^{m \times d}$, $y_{\text{pub}} \in \mathbb{R}^m$
**Require:** Parameter vectors $\beta^+, \beta^- \in \mathbb{R}^d$
**Require:** Loss function $\ell(\beta)$
**Require:** Embedding function $\phi$
**Ensure:** Reconstructed sample $\tilde{x}$
  Estimate the Hessian $\hat{H}$ using Eq. (13)
  Reconstruct the embedding $\tilde{z}$ using Eq. (12)
  **if** $\phi(x) = x$ **then**
    Directly recover $\tilde{x} = \tilde{z}$
  **else**
    Reconstruct the input $\tilde{x}$ using Eq. (6)
  **end if**
  Return $\tilde{x}$

---

### 3.3 Multiclass Classification and Label Inference

In a multiclass classification setting, our approach described in Eq. (12) facilitates an estimation of parameter gradients with respect to the deleted sample. Employing Eq. (6), initially defined in Section 3.1, allows for reconstructing the deleted data point. In models using linear layers directly post-embedding, recovery becomes straightforward. However, for models with multiple class-specific parameters, label inference requires additional steps.

Employing a softmax nonlinearity for outputting probability vectors, we observe that the derivative of the loss with respect to the bias for the correct label is distinctively negative, setting it apart from the other biases which demonstrate positive derivatives under typical loss functions:

$$\nabla_{b_j}[-\ln f_y(x; \beta^+)] = f_j(x; \beta^+) - \mathbf{1}[j = y]. \tag{14}$$

The deleted label $y$ can then be inferred as:

$$\hat{y} = \arg\min_j \nabla_{b_j} \ell(\beta; x, y)\big|_{\beta = \beta^+}. \tag{15}$$

This approach significantly extends the capabilities of privacy attacks to encompass a wider array of multiclass classification models.

## 4 Experiments

We assess our attack across diverse datasets, including tabular and image data, and for both classification and regression tasks. Initially, we train a model on the complete dataset $X_{\text{priv}}, y_{\text{priv}}$ to derive parameters $\beta^+$, and then retrain it—excluding a single sample $(x, y)$—to obtain $\beta^-$. We note that $\beta^-$ is achieved through full retraining, not approximate unlearning methods—our attack does not depend on imperfect unlearning to be effective.

Our attack leverages public data samples from the same distribution as the training data but does not require knowledge of the deleted sample's features or label. We evaluate our approach against two baselines that utilize public data:

"**Avg**": Predicts the deleted sample as $\hat{x} = \frac{1}{m} \sum_{x \in X_{\text{pub}}} x$, an average of the public samples.

"**MaxDiff**": Identifies the public sample that maximizes the prediction discrepancy between $\beta^+$ and $\beta^-$ as $\hat{x} = \arg\max_{x \in X_{\text{pub}}} ||x^T(\beta^+ - \beta^-)||$.

These baselines exploit similarities between public and private data, with "MaxDiff" additionally considering the change in model parameters. Our threat model differs significantly from that of Salem et al. (2020), who assume black-box access and a simpler model update scenario; details and comparisons are found in Appendix B.

For practical simulations, each dataset is split into two: one for private training and the other for public samples. We simulate the deletion of each sample in the private set, retrain the model, and attempt its reconstruction using our method and the baselines.

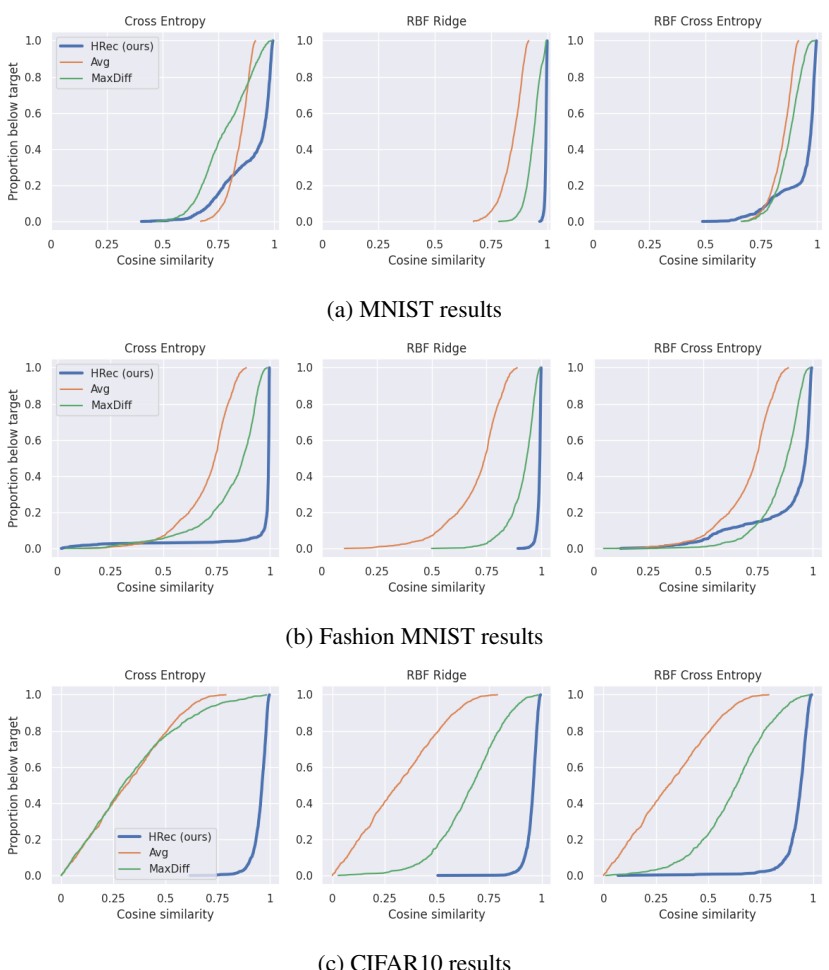

Figure 3: Cumulative distribution function of cosine similarity between deleted and reconstructed sample via the average, MaxDiff, and HRec (our) attack on MNIST, Fashion MNIST and CIFAR10 for three model architectures (linear cross-entropy, ridge regression over $4096$ random Fourier features, and cross-entropy over $4096$ random Fourier features). **Lower curves correspond to more effective attacks than higher curves. Our attack achieves better cosine similarity with the deleted sample across all settings**; the effect is especially apparent in the denser CIFAR10 dataset.

Hyperparameters, specifically the $\ell_2$ regularization strength $\lambda$, are optimized on the private set and remain constant when recalculating $\beta^-$. We explore attacks on unregularized models in Appendix C.

We quatify the efficacy of our attacks and baselines with the cosine similarity between deleted and reconstructed samples, aggregated into a Cumulative Distribution Function (CDF) of these similarity scores; a sharp peak near 1 indicates precise reconstruction. **CDFs which are "below" others correspond to more effective attacks**, which have a higher fraction of reconstructed points with very high similarity to the original data. The performance of target models is evaluated in Appendix D.

## 4.1 Image Data

We evaluate our generalized attack methodology, outlined in Algorithm 1, across three distinct model configurations on Fashion MNIST (FMNIST), MNIST, and CIFAR10 datasets. Fashion MNIST and MNIST consist of $28 \times 28$ grayscale images, whereas CIFAR10 includes $32 \times 32 \times 3$ RGB images, with all datasets aimed at 10-way classification (Xiao et al., 2017; LeCun et al., 1998; Krizhevsky et al., 2009). Each dataset undergoes a normalization process where input features are scaled to the range $[-1, 1]$.

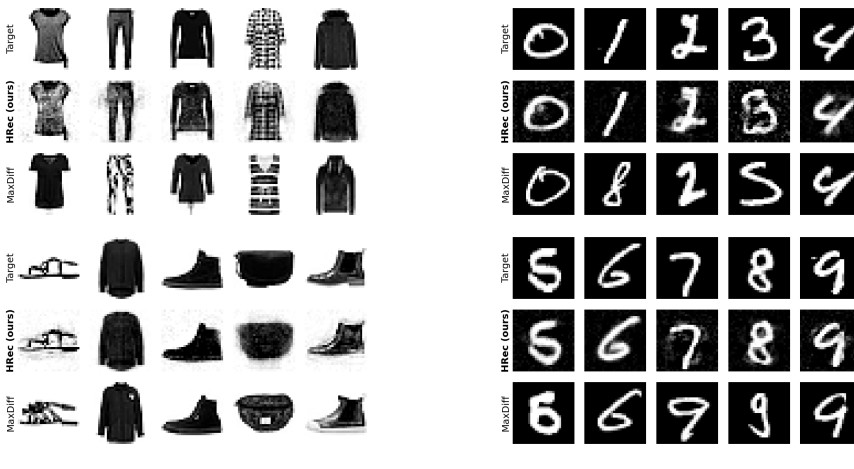

|                |                |
| :------------: | :------------: |
| (a) Fashion MNIST | (b) MNIST |

Figure 4: Sample reconstructions on Fashion MNIST/ MNIST for a $40K$ parameter model (cross-entropy over random Fourier features of the raw input). We randomly chose one deleted sample per label (Rows 1, 4) and compared them against the reconstructed sample using our method (HRec, Rows 2, 5) and a perturbation baseline (MaxDiff, Rows 3, 6) which searches for the public sample with the largest prediction difference before and after sample deletion. HRec produces reconstructions that are highly similar to the deleted images.

**Cross-entropy Loss for Multiclass Classification:** We first consider a linear model with a softmax output, trained to minimize the cross-entropy loss $\ell_{CE}(\beta, x, y) = -\log \sigma_y(x^\top \beta) + \lambda\|\beta\|_2^2$. This model facilitates direct gradient estimation with respect to the parameters $\beta^+$ proportional to the raw input $x$, negating the need for Eq. (6)

**Ridge Regression with Random Fourier Features:** The second model incorporates an embedding function $\phi$, generating random Fourier features (Rahimi & Recht, 2007). The associated loss function, $\ell_{Ridge}(\beta, \phi(x), y) = \|\phi(x)^\top \beta - y\|_2^2 + \lambda\|\beta\|_2^2$, admits an analytical solution for the Hessian matrix $H = \phi(X)^\top \phi(X)$ concerning $\phi(x)$, but requires embedding inversion via Eq. (6).

**Cross-entropy Loss with Random Fourier Features:** This model merges the complexities of the first two: it lacks a closed-form update solution and necessitates embedding inversion, addressing scenarios with both softmax nonlinearity and random Fourier embeddings.

Figure 3 illustrates the efficacy of our attack across all three model types on MNIST, FMNIST and CIFAR10, demonstrating the capability to consistently recover samples highly similar to the original, deleted samples. Figures 2, 4a, and 4b depict randomly sampled deletions and their nearest recovered samples using HRec and MaxDiff techniques, particularly under the challenging conditions of cross-entropy loss over random Fourier features. Further results for additional model configurations can be found in Section E, expanding upon the robustness and versatility of our attack methodology across varied settings and data modalities.

## 4.2 Tabular Data

**Ridge Regression for Income Prediction** We perform an attack on ridge regression models using American Community Survey (ACS) income data from 2018 (Ding et al., 2021). Our approach involves direct application of Eq. (4) along with an intercept normalization technique. Figure 5 (first row) illustrates the attack performance. If the covariance matrix of the private data and the regularization parameter $\lambda$ were known, our approach could perfectly recover the deleted sample. However, our approximation assumes $\lambda = 0$ and estimates the covariance matrix from public samples, which introduces some estimation error. Despite these approximations, we achieve near-perfect reconstruction of the deleted sample.

**Ridge Regression with Random Features** In a variation of the previous attack, we target a ridge regression model trained with random Fourier features (Rahimi & Recht, 2007). Assuming access to both the random Fourier features and the model weights, we reconstruct the embedding $\tilde{z}$ of the deleted sample and then solve an inverse problem to recover the original features. The results,

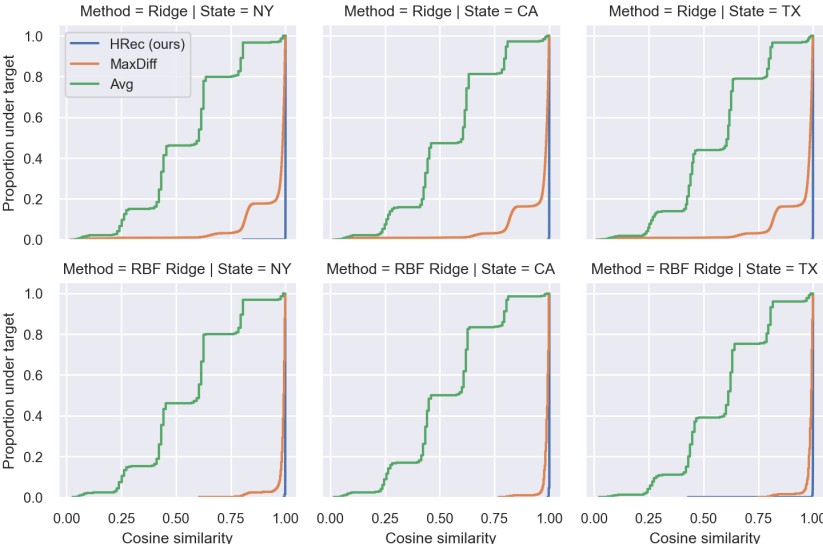

Figure 5: ACS Income Regression. Target models are ridge regression with tuned hyperparameters on original features (first row), and over random Fourier features (second row). ACS Income data from three states are used to demonstrate the effectiveness of our attack. Given the analytical single-sample update rules of linear regression, our attack (HRec) reconstructs the deleted sample almost perfectly on all datasets and different embedding functions.

depicted in the second row of Figure 5, demonstrate significant improvement over the baselines, achieving almost perfect reconstruction accuracy.

**Binary Classification for Income Level Prediction** We extend our analysis to binary classification tasks using logistic regression and support vector machines (SVMs) with squared hinge loss. Both models allow analytical computation of their Hessian matrices:

$$\hat{H} = X_{\text{pub}}^{\top} D X_{\text{pub}},$$

where $D \in \mathbb{R}^{M \times M}$ is a diagonal matrix. For logistic regression, the diagonal terms are defined as $D_{ii} = \sigma(x_i^{\top}\beta^+)(1 - \sigma(x_i^{\top}\beta^+))$, with $\sigma$ being the sigmoid function. For SVMs, the terms are $D_{ii} = \mathbb{1}(1 - y_i x_i^{\top}\beta^+ \geq 0)$. The reconstruction of the deleted sample's features is obtained via $\hat{H}(\beta^+ - \beta^-)$, enhanced by the intercept normalization trick. The performance of these attacks is showcased in Figure 6a, where our method, HRec, consistently outperforms the baselines across all datasets and model classes.

**Binary Classification with Random Features** Lastly, we attack binary classification models trained on an enriched set of random Fourier features. Figure 6b presents the performance curves for our attack compared to various baselines. Our method outperforms all baselines in attacks on logistic regression models and performs competitively with the MaxDiff baseline in attacks on SVM models, highlighting the efficacy of our approach.

Our comprehensive attack strategies on regression and binary classification models demonstrate the potential vulnerabilities in these machine learning setups, especially when certain model parameters or features are accessible to an adversary. These findings underscore the need for robust privacy-preserving mechanisms in machine learning applications.

## 5 Conclusion

We present a strong reconstruction attack that targets data points that are deleted from simple models. Our reconstructions are nearly perfect for linear regression models, and still achieve high-quality reconstructions for linear models constructed on top of embeddings, and for models which optimize various objective functions. This shines a light on the privacy risk inherent in even very simple

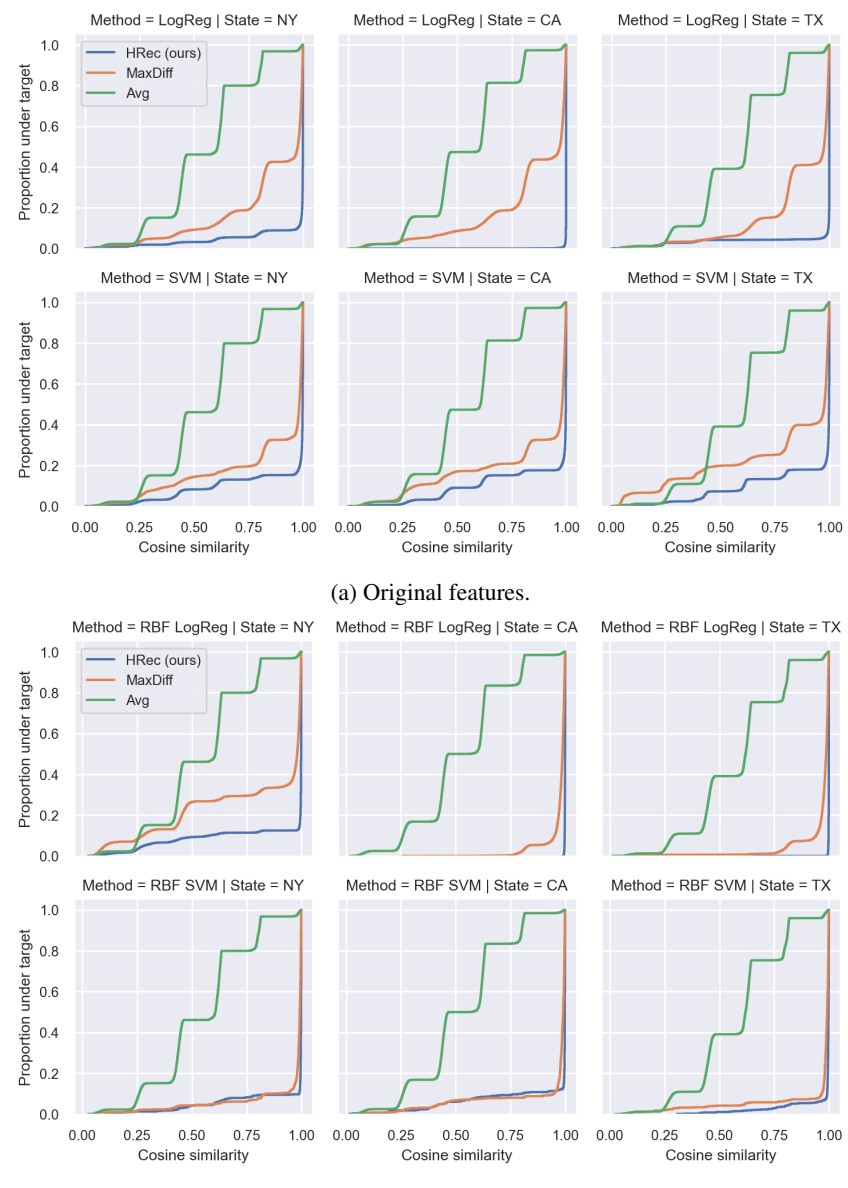

(a) Original features.

(b) Random Fourier Features.

Figure 6: ACS Income Level Prediction. Target models are logistic regression (first row) and SVM (second row) for binary classification tasks. These methods do not have analytical forms of the single-sample update; however, our approximation using Newton's update facilitates the outstanding performance of HRec among all attacks. While this reconstruction is imperfect due to approximation errors, a large number of deleted samples can still be reconstructed with high similarity scores.

models in the context of data deletion or "machine unlearning", and motivates using technologies like differential privacy to mitigate reconstruction risk.

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

## A  Membership Inference Attacks on Linear Models

Linear models usually have lower privacy risks compared to neural networks because the parameters of a linear model are significantly fewer. To demonstrate the low privacy risks, we conduct a state-of-the-art membership inference attack (MIA) proposed by Carlini et al. (2022) — Likelihood Ratio Attack (LiRA) — on the same tabular tasks. The goal of MIA is to determine whether a sample is in the training set of the target model.

On each task, we split the dataset into three splits, including 40% for training the target model, another 40% as the public samples for learning shadow models, and the rest 20% as the holdout set for evaluation. After training the target model on the private training set, we train 64 shadow models using the same optimization algorithm on the public samples with Bootstrap. Then, we use the joint set of the private samples and the holdout samples as samples under attack, and evaluate the attack performance.

As shown in Figure 7, the attack performance is close to random guessing, which implies that it is already challenging to determine which sample has been used in training when the target model is linear.

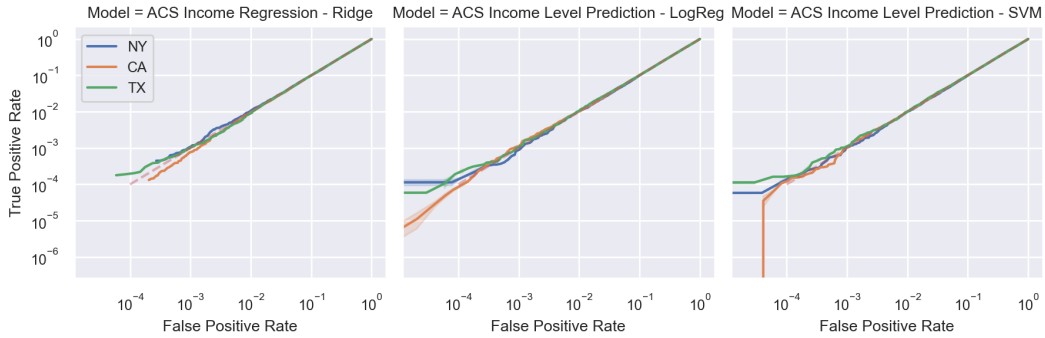

Figure 7: Membership inference attacks on ACS tasks.

## B  Additional Comparisons

Here we provide a limited comparison of Updates-leak Salem et al. (2020) against our method and baselines for the same simple model architecture (cross-entropy loss over a linear model on top of 4096 random Fourier features). We stress that the threat model for Updates-Leak differs from our own in two important ways. First, they assume query access to the model, while we assume access to the parameters. Second, we carry out our attack on two models, fully trained to convergence on two different datasets, $((X_{\text{priv}}, y_{\text{priv}})$ and $(X_{\text{priv}} \setminus x, y_{\text{priv}} \setminus y))$. In contrast, Updates-leak instead attacks the difference between a model trained on $(X_{\text{priv}} \setminus x, y_{\text{priv}} \setminus y)$ and an updated model where in the update, only a single gradient descent step is taken on the 'update' sample $(x, y)$ (single sample attack version). The Updates-Leak approach also incurs a significantly higher computational cost due to its shadow model and encoder learning approach. For these reasons, we limit the comparison to a single model architecture on CIFAR10 while stressing this comparison is not 'apples to apples'. In particular, even though we plot the reconstrution cosine similarity curves on the same axis (and see that ours improves), our technique and UpdatesLeak are attacking *different pairs of models* (we attack the model that results from full retraining, whereas they attack the model that results from a single gradient update).

Figure 8 shows the cosine similarity comparison, and Figure 9 show some example reconstructions for Updates-Leak in this scenario. We leveraged their publicly available code to produce these comparisons, using their default configuration (10,000 shadow models are used for training, and their DC-GAN generator is trained for 10,000 epochs on the shadow model dataset).

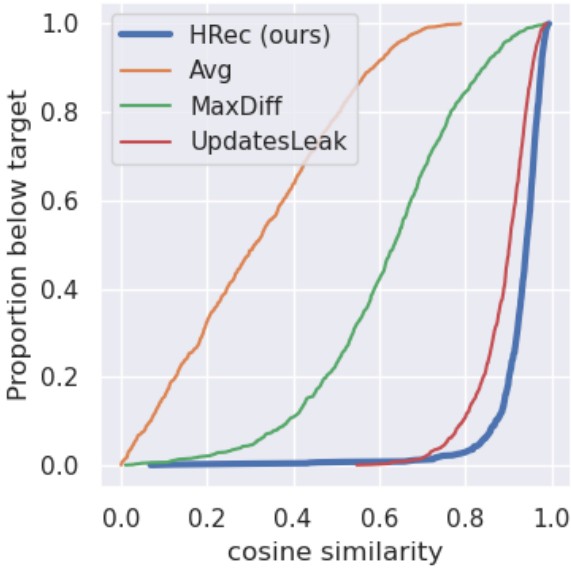

Figure 8: Cumulative distribution function of cosine similarity between the target (deleted) sample and the reconstructed sample via the average, MaxDiff, Updates-Leak, and HRec (our) attack on CIFAR10 on a simple model (cross-entropy loss over a linear model on top of $4096$ random Fourier features). **All attacks save for Updates-Leak operate against the full retraining baseline, that is the comparison between two models trained from scratch until convergence in a dataset with and without the 'deleted' sample $((X_{\mathbf{priv}}, y_{\mathbf{priv}})$ and $(X_{\mathbf{priv}} \setminus x, y_{\mathbf{priv}} \setminus y))$, Updates-Leak instead attacks two models, one trained until convergence on the dataset without the sample $((X_{\mathbf{priv}} \setminus x, y_{\mathbf{priv}} \setminus y))$, and one that took a single gradient descent step on the loss of the updated sample $x, y$.** Here lower curves dominate higher curves. Our attack achieves better cosine similarity with the deleted sample.

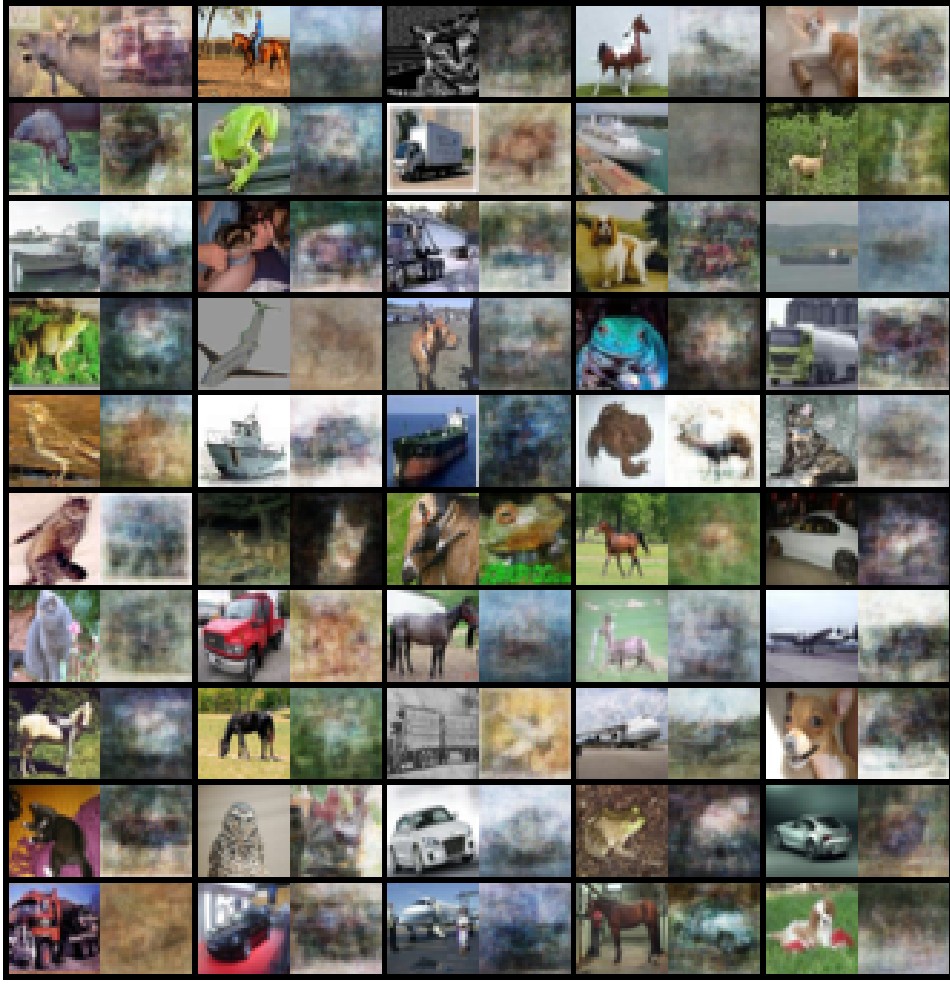

Figure 9: Sample reconstructions for Updates-Leak on CIFAR10. Original images and their corresponding reconstruction are shown side by side in an alternating fashion. The model architecture is cross-entropy loss over a linear model on top of $4096$ random Fourier features. The models before and after the update differ in a single gradient descent step being taken on the update sample $(x, y)$.

## C  Attacking Unregularized Models

In our main experiments, we emulate the more realistic situation where the model maintainer tunes the hyperparameter of the target model on the entire dataset, and keeps it fixed during unlearning. Since the impact of regularization on the attack performance is rather challenging to analyze and not immediately obvious, we here present results on attacking models without regularization.

### C.1  ACS Income Regression

On this task, the model maintainer directly optimizes the following objective without the regularization term:

$$\beta^* = \arg\min_{\beta} \|X\beta - y\|_2^2 \tag{16}$$

This problem admits an analytical expression for $\beta^+$ and $\beta^-$, which can be written as:

$$\beta^+ = C^{-1} X_{\text{priv}}^\top y_{\text{priv}}, \tag{17}$$

$$\beta^- = (C - xx^\top)^{-1}(X_{\text{priv}}^\top y_{\text{priv}} - x^\top y), \tag{18}$$

where $C = X_{\text{priv}}^\top X_{\text{priv}}$ is the covariance matrix. In the scenario where the inverse of the covariance matrix doesn't exist, we use the Moore–Penrose inverse instead. Our attack still stays the same.

The results are presented in Figure 10, and we can see that without regularization

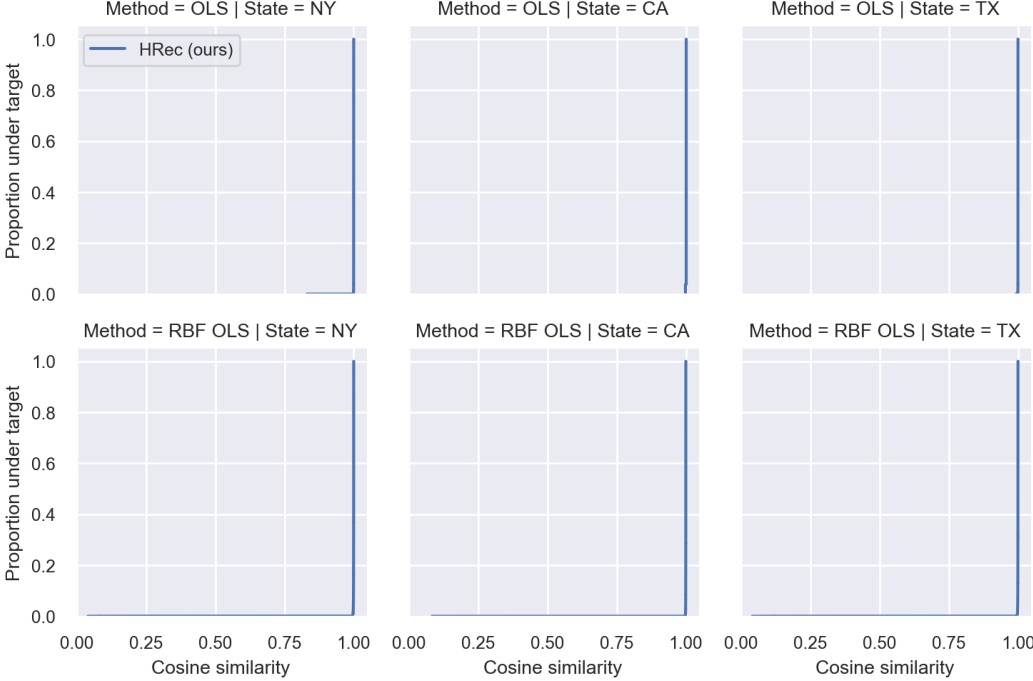

Figure 10: ACS Income Regression. Target models are ordinary linear regression on original features (first row), and over random Fourier features (second row). Our attack HRec reconstructs the deleted sample almost perfectly.

### C.2  Image Classification Tasks

Here we additionally show results across CIFAR10, MNIST, and Fashion MNIST on the more challenging target model scenario (RBF Cross Entropy). For these results, the model maintainer does not use any form of regularization. Results are shown in Figure 11.

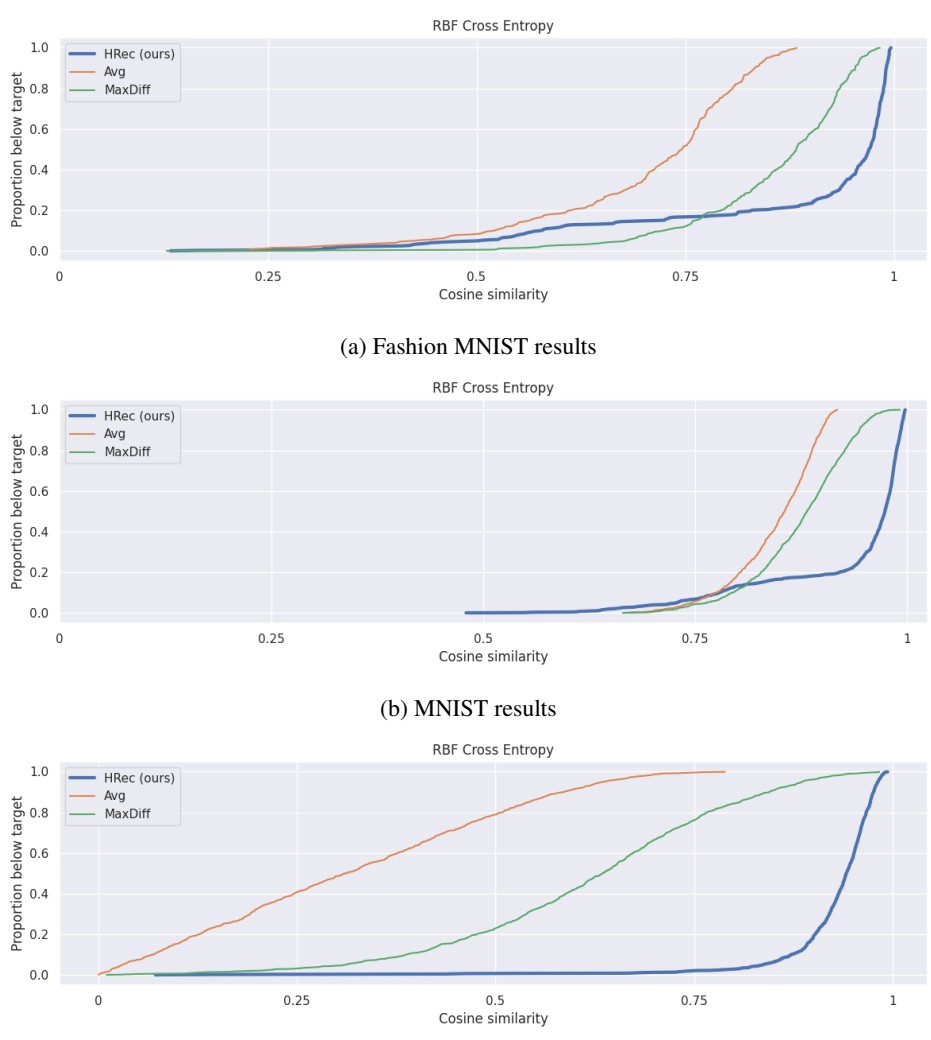

(a) Fashion MNIST results

(b) MNIST results

(c) CIFAR10 results

Figure 11: Cumulative distribution function of cosine similarity between the target (deleted) sample and the reconstructed sample via the average, MaxDiff, and HRec (our) attack on Fashion MNIST, MNIST, and CIFAR10 for a target model using cross-entropy over 4096 random Fourier features. In this scenario, the model maintainer does not use any form of regularization when training the original or updated model. Here lower curves dominate higher curves. Our attack achieves better cosine similarity with the deleted sample across all settings; the effect is especially apparent in the denser CIFAR10 dataset.

# D   Performance of Target Models

## D.1   ACS Income Tasks

Table 1: Performance of target models on ACS Income tasks. Regression tasks are evaluated using $r^2$, which indicates the portion of explained variance, and classification tasks are evaluated using $F1$ score since class labels are not balanced.

| STATE | | NY | | CA | | TX | |
|---|---|---|---|---|---|---|---|
| TASK | TARGET MODEL | | +RBF | | +RBF | | +RBF |
| REGRESSION (R2) | RIDGE REGRESSION | 0.2755 | 0.32712 | 0.30139 | 0.3510 | 0.3089 | 0.3510 |
| CLASSIFICATION (F1) | LOGISTIC REGRESSION | 0.7154 | 0.7230 | 0.7313 | 0.7413 | 0.6889 | 0.7009 |
| | LINEAR SVM | 0.7099 | 0.7149 | 0.7345 | 0.7325 | 0.6868 | 0.6968 |

## D.2   Image tasks

Table 2: Out of sample accuracy of target models on Image tasks

| DATASET | LINEAR CROSS-ENTROPY | RBF RIDGE | RBF CROSS ENTROPY |
|---|---|---|---|
| CIFAR10 | 39.5% | 48.7% | 50.4% |
| MNIST | 91.6% | 96.2% | 96.4% |
| FASHION MNIST | 84.5% | 87.3% | 88.4% |

# E  Additional Results on Image Datasets

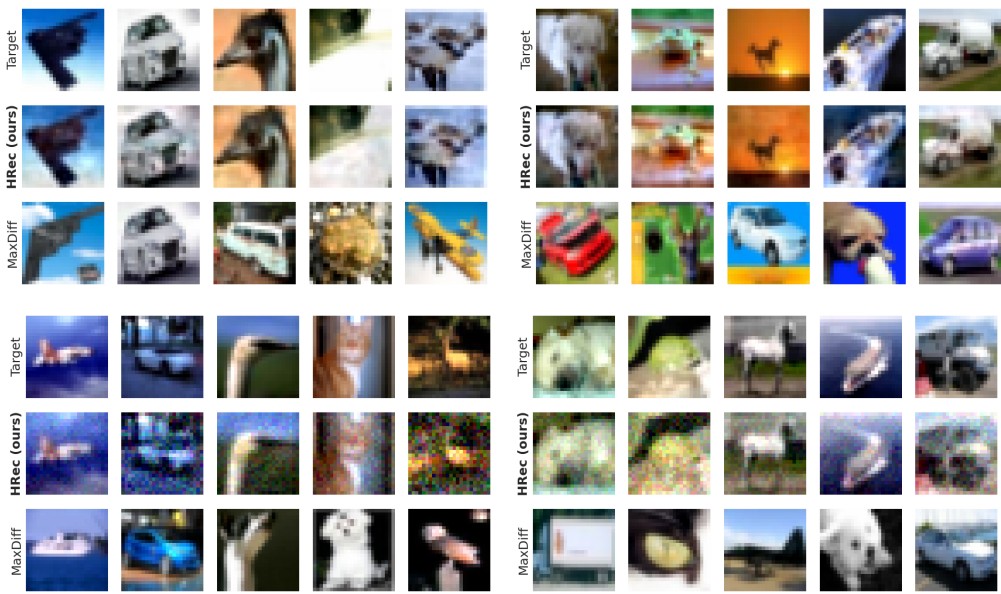

Figure 12: Sample reconstructions on CIFAR10. Rows 1-3 rows show results of attacking a linear cross-entropy model, and rows 4-6 show similar results for ridge regression over 4096 random Fourier features. We randomly chose one deleted sample per label (shown in rows 1 and 4) and compared them against the reconstructed sample using our method (HRec, rows 2 and 5) and a perturbation baseline (MaxDiff, rows 3 and 6) which searches for the public sample with the largest prediction difference before and after sample deletion. HRec produces reconstructions that are highly similar to the deleted images.

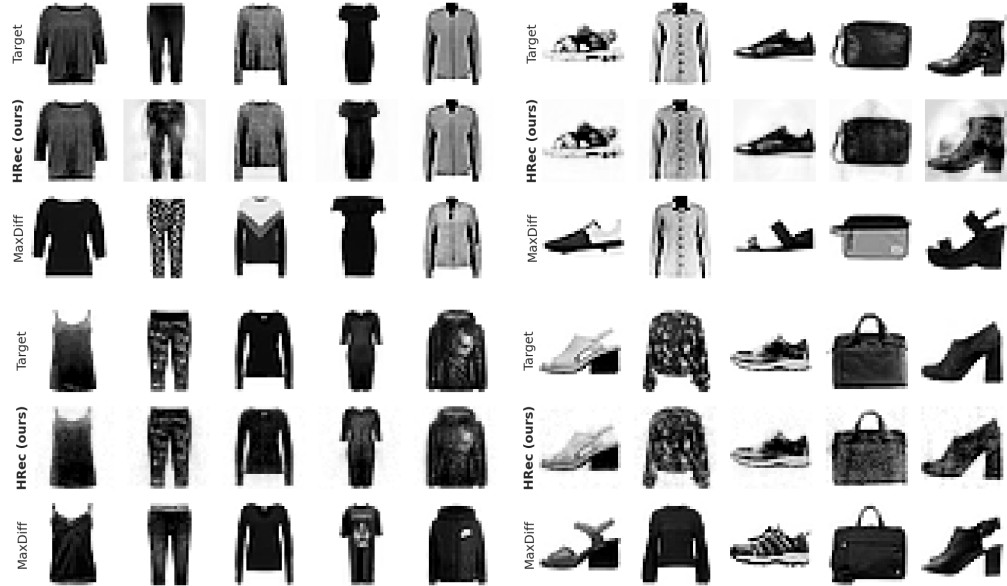

Figure 13: Sample reconstructions on Fashion MNIST. Rows 1-3 rows show results of attacking a linear cross-entropy model, and rows 4-6 show similar results for ridge regression over $4096$ random Fourier features. We randomly chose one deleted sample per label (shown in rows 1 and 4) and compared them against the reconstructed sample using our method (HRec, rows 2 and 5) and a perturbation baseline (MaxDiff, rows 3 and 6) which searches for the public sample with the largest prediction difference before and after sample deletion. HRec produces reconstructions that are highly similar to the deleted images.

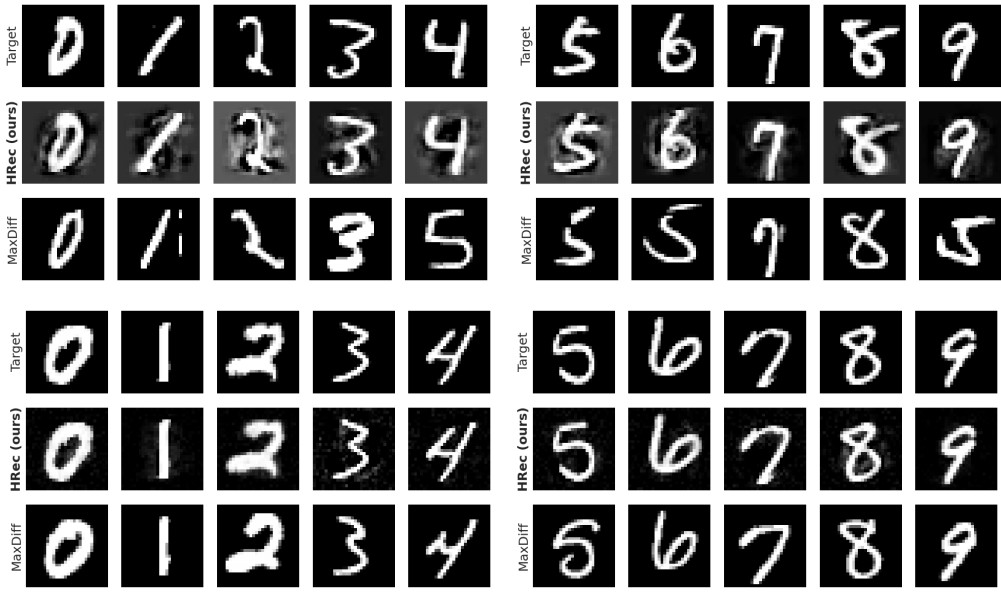

Figure 14: Sample reconstructions on MNIST. Rows 1-3 rows show results of attacking a linear cross-entropy model, and rows 4-6 show similar results for ridge regression over $4096$ random Fourier features. We randomly chose one deleted sample per label (shown in rows 1 and 4) and compared them against the reconstructed sample using our method (HRec, rows 2 and 5) and a perturbation baseline (MaxDiff, rows 3 and 6) which searches for the public sample with the largest prediction difference before and after sample deletion. HRec produces reconstructions that are highly similar to the deleted images.

