# OpenReview forum: "Reconstruction Attacks on Machine Unlearning: Simple Models are Vulnerable"
_NeurIPS.cc/2024/Conference — NeurIPS 2024 poster_

### Official Review · Reviewer_zg9a · 2024-07-11

**Soundness:** 2
**Presentation:** 3
**Contribution:** 2
**Rating:** 4
**Confidence:** 4

**Summary:**

This paper studies reconstruction attacks on machine unlearning. The authors propose a reconstruction attack that can accurately recovers the deleted sample given the pair of linear models before and after sample deletion. This is made possible by leveraging the closed-form single-sample training algorithm for linear regression as well as the ability to accurately estimate the covariance matrix of the training data from a public dataset. They also extend the attack to the setting where the model consists of a (fixed and known) embedding function, followed by a trained linear layer.

**Strengths:**

- The problem studied in this paper is interesting.

- The proposed attack achieves good performance for linear regression models.

**Weaknesses:**

- The proposed attack is based on strong assumptions. This paper assumes that the attacker has knowledge of the training data distribution, the model’s loss function, and the embedding function. However, this information is usually not published by the model maintainer. It is unclear how the attacker can obtain this information in practice. The authors should provide more details to justify these assumptions.

- The proposed attack has limited applications. This paper mainly focuses on linear models, and it is unclear whether the proposed attack can maintain good performance with more complex models.

- It is unclear which unlearning method is adopted in the experiments. Will the attack performance vary when different unlearning methods are applied?

- The proposed attack relies on a public dataset with the same distribution as the training data. The authors do not provide information about the size of this dataset in their experiments. It would be more convincing if they could evaluate the effect of dataset size on the attack performance.

**Questions:**

See above weaknesses.

**Limitations:**

Yes

---

> ### Author Rebuttal · Authors · 2024-08-06
>
> *[Response to Weaknesses 1]*
>
> The assumption of having access to the training data distribution is standard throughout the membership inference literature. Because we are trying to compare the risks of data deletion to the known risks absent data deletion, we adopt the same model. In general we view this as an appropriately cautious assumption about what an attacker might know, especially when attacks are being used to assess model risk. You are right that it might sometimes be difficult for an attacker to find an appropriate sampling distribution, and understanding how much it can be relaxed is an important question in general for the entire literature on privacy attacks; but one that we view as outside the scope of this paper.
>
> In our experiments, we assume the most commonly-used loss functions, including MSE, logistic loss, and hinge loss, and show positive results for each. Random fourier features are also widely used in areas where latency is a concern.
>
> ----
>
> *[Response to Weaknesses 2&3]*
>
> An important aspect of existing machine unlearning approaches is that almost all of them aim to approximate full retraining with reduced computational cost. These approximations are proposed due to the intense compute required for full retraining, especially for large neural networks and LLMs.
>
> Our study is explicitly focused on exposing the risk present in even very simple models, such as linear regression, logistic regression, SVMs, and feature augmentation using random Fourier features. For such models, full retraining is feasible, and would be the expected solution to the data deletion problem, as computational approximations to retraining are not needed.
>
> Full retraining for a single data deletion in linear regression is equivalent to updating the model parameters using Newton's update as we discussed in Sec. 3.2. By leveraging this concept, our attack achieves almost perfect reconstruction, and the only error source is the estimation of the Hessian matrix using public data. For more complex models, a number of popular unlearning methods approximate full retraining by taking a Newton update. When we apply our attack to more complex models, we act as if full retraining results in the model that is derived from taking a newton step --- and the reason our reconstruction performance degrades is simply that this approximation becomes imperfect. However, if rather than full retraining, the machine unlearning method employed was one that simply took a newton step, our reconstruction would again be near perfect. We will elaborate on this point in the revision.
>
> ----
>
> *[Response to Weaknesses 4]*
>
> As we mention in Sec. 4, in all our experiments, the dataset is split into two halves; the first half is used for training the private model, and the second half is used for attacking the trained model.
>
> The public dataset is used to estimate the Hessian matrix, which is the covariance matrix for linear regression. The quality of the estimation increases asymptotically with respect to the number of samples in the public dataset, and it is the only source of error in attacking linear regression, thus, the attack performance behaves similarly to the quality of the estimation. This grows with the dimension of the model; we can elaborate on this point in the revision.

---

### Official Review · Reviewer_YZsT · 2024-07-12

**Soundness:** 4
**Presentation:** 4
**Contribution:** 2
**Rating:** 6
**Confidence:** 5

**Summary:**

This paper presents reconstruction attacks against Machine Unlearning in the following sense: the attacker is assumed to have access to a model's parameters before and after the removal of a single data point; they then produce a guess for this point, which is evaluated in terms of its cosine similarity to the original data point.

**Strengths:**

This paper is nice and easy to read. The description of the attacks is easy to follow, and the theoretical derivations are interesting.
The specific application of reconstruction attacks against machine unlearning is, to my knowledge, novel; it stems from a large body studying the privacy of machine unlearning.
The presented problem is also well-scoped, and this research opens the space for new studies in the area.

**Weaknesses:**

1. The threat model is unrealistic: it is far fetched to assume that the attacker has access to the model parameters (before and after unlearning), and yet at the same time to assume that they cannot see the target point x in this process. Now, strong assumptions such as this one have been used in prior literature; but usually their purpose is to set upper bounds on the adversary when proving the security of defenses. That is not the case here.
A second assumption that is quite strange is that the attacker somehow knows the model's parameters, yet they don't know the training set Xpriv, and they need to rely on a public one.

2. The authors focus on the very limited and quite simplistic scenario specified above. Yet they had various options for exploration:
- what if the attacker only has black-box access to the model? Based on similar prior work that evaluated both white- and black-box access (e.g., Balle et al.), I would expect your attacks to transfer well.
- You mentioned DP in several places, yet provided no evaluation of said defense: what parameter set can prevent these attacks?
- What if more than 1 points were unlearned at once? Would your attack apply?

**Questions:**

Can you please explain your threat model choices (see Weaknesses above)?

Typo: "To simply the notation"

Baseline: an interesting baseline to consider would be the point from the public set that is closest to the target point in the private set; this would intuitively be a better baseline than MaxDiff. Could your methods beat this baseline?

**Limitations:**

These were appropriately discussed, although the authors should better emphasize that the threat model is not realistic.

---

> ### Author Rebuttal · Authors · 2024-08-06
>
> *[Response to Weaknesses]*
>
> We take the position that security assurances should be based on minimal assumptions. Here, we view the assumption that an attacker who has API access to the model does -not- have access to model parameters to be dangerously strong. Consider for example a d dimensional linear model, as we study initially in our work. Just from query access, an attacker can recover the model parameters by querying the model on d linearly independent points and solving a system of linear equations. This requires no knowledge of the data distribution. So, there is no meaningful difference between white box and black box access in such scenarios.
>
> What about for more complex models? Even there the boundary between white-box and black-box access has been blurring. For example recent work [1] has shown that it is possible to reconstruct the embedding matrix from a production LLM model with only black-box access. And of course, open source models explicitly release parameters with each version of the model.
>
> As for the fact that the attacker does not know the training set: first we note that the goal of a reconstruction attack is to recover data from the training set. Even for linear models, in which recovering the parameters given only black box access to the model is trivial, it is in general not possible to recover the training set from the parameters of a single linear model. To see this note that many datasets produce the same set of parameters --- linear regression parameters are e.g. invariant to rotations of the dataset For linear models, and the model can be expressed with only O(d) bits of information (after discretizing weights), whereas a dataset with n datapoints and d features requires Omega(n * d) bits to represent.  For larger models, existing state of the art methods such as [2,3] rely on creating a set of samples such that the realized model parameters satisfy the KKT conditions of the loss function (the parameters minimize the loss at recovered samples). The success of these approaches requires that datasets are trained on a small number of samples (e.g. 500).
>
> [1] Carlini, Nicholas, et al. "Stealing part of a production language model." arXiv preprint arXiv:2403.06634 (2024).
>
> [2] Haim, Niv, et al. "Reconstructing training data from trained neural networks." Advances in Neural Information Processing Systems 35 (2022): 22911-22924.
>
> [3] Buzaglo, Gon, et al. "Deconstructing data reconstruction: Multiclass, weight decay and general losses." Advances in Neural Information Processing Systems 36 (2024).
>
> ----
>
> Re: multi-sample deletion, our approach recover an approximation of the gradient sum for all the deleted samples. A determined adversary can poison an unlearning round by requesting deletion of n-1 samples known to the adversary, and so can recover the gradient of the single unknown point from this sum. Nevertheless we agree that multi-sample recovery is an important question that our work does not address, and think this is one of the most interesting questions for future work arising from our paper.
>
> On defenses: differential privacy guarantees compose, so given two models (from before and after a deletion) each trained with $\epsilon$-differential privacy, we have the guarantees of $2\epsilon$-differential privacy. When examples are drawn from a distribution uniform on some data domain $\mathcal{X}$, then $\epsilon \leq \Omega(\log|\mathcal{X}|) is enough to provably prevent reconstruction. More generally differential privacy bounds the advantage that an adversary has attempting to reconstruct a sample given the model parameters (compared to their success rate ``just guessing''), and so precise guarantees depend on the entropy of the data distribution (as low entropy distributions allow high rates of "reconstruction" without the adversary even needing to see the model. These are standard/generic properties of differential privacy which is why we did not devote space to it, but we are happy to elaborate in the revision.
>
> ----
>
> *[Response to Questions]*
>
> Unless we misunderstand, the ''baseline'' you propose could not be implemented without already knowing the private dataset: otherwise the attacker would have no way of knowing what the ``closest point to the target point'' in the private set is. Of course with knowledge of the private training set, there is nothing left to do. If we misunderstand your proposal please let us know, and we'll be happy to discuss further!

---

> > ### Comment · Reviewer_YZsT · 2024-08-12
> >
> > Thank you for your response, and in particular for addressing my comments on:
> > - threat model: agreed on the linear model, and if I understand correctly also to the "Fixed Embedding Functions" setting. I'm on the fence as to whether this threat model is any useful for more general models, but I take your point.
> > - multi-sample: noted, although It seems to me that this could've easily featured in one of your experiments.
> > - DP: this simple derivation is actually quite interesting to me personally. "These are standard/generic properties of differential privacy which is why we did not devote space to it, but we are happy to elaborate in the revision.": it's of course entirely your call on whether to include them or not in the paper.
> >
> > Regarding the "baseline": I meant that, as an evaluation baseline, you could consider an (optimal, to some extent) adversary, who outputs the point from the public dataset that is closest to the target. Of course, this would be unrealistic as an attacker; however, to my understanding, it should provide a good intuition as to how much better than "just using the public data" your method is doing.
> > This is not a requirement for acceptance; just a mere suggestion.
> >
> > My (positive) score is unchanged.

---

> > > ### Author Response · Authors · 2024-08-12
> > > **Thanks**
> > >
> > > Thanks for the engagement, the helpful suggestions, and the positive score. We appreciate it!

---

### Official Review · Reviewer_VcL6 · 2024-07-13

**Soundness:** 3
**Presentation:** 2
**Contribution:** 3
**Rating:** 6
**Confidence:** 3

**Summary:**

The authors propose an attack that can accurately recover unlearned samples through reconstruction attacks on linear regression models. They extend this work to include linear models with fixed embeddings and generalize it to more generic loss functions and model architectures by employing Newton’s method for the reconstruction attack. This work significantly contributes to understanding the privacy vulnerabilities in machine unlearning.

**Strengths:**

1. The authors provide rigorous theoretical proof of the reconstruction attack.
2. They conducted thorough experiments across different tasks, datasets, and architectures, demonstrating the effectiveness of their attack.
3. The extension of the work to linear models with fixed embeddings and the generalization to other loss functions and model architectures showcase the adaptability and robustness of their method.

**Weaknesses:**

1. This work is limited to the exact unlearning scenario, i.e. retraining from scratch without the unlearned data, and focuses solely on unlearning a single data point. In contrast, the scenarios that have received more attention in unlearning research involve approximate unlearning and unlearning multiple data points at the same time.
2. The experimental evaluation lacks diversity in metrics, which could provide a more comprehensive understanding of the attack's effectiveness.
3. There is limited discussion on the potential defenses against the proposed attack.

**Questions:**

1. Is “Avg” a valid or commonly used baseline? It seems too straightforward to predict the deleted example as an average of the public samples, as described in Section 4. Can the authors elaborate on the principle or intuition behind this choice?
2. How robust is this attack across different configurations? For example, how does the attack's performance vary when using different model architectures on the same dataset?
3. Can the authors elaborate on the computational complexity and scalability of the proposed attack method when applied to various datasets and model architectures?
4. What practical countermeasures can be implemented to mitigate the identified privacy vulnerabilities (e.g. the proposed reconstruction attack in this paper) in machine unlearning?

**Limitations:**

See Weaknesses Section.

---

> ### Author Rebuttal · Authors · 2024-08-06
>
> *[Response to Weaknesses]*
>
> Our study is explicitly focused on exposing the risk present in even very simple models, such as linear regression, logistic regression, SVMs, and feature augmentation using random Fourier features. For such models, full retraining is feasible, and would be the expected solution to the data deletion problem, as computational approximations to retraining are not needed.
>
> Full retraining for a single data deletion in linear regression is equivalent to updating the model parameters using Newton's update as we discussed in Sec. 3.2. By leveraging this concept, our attack achieves almost perfect reconstruction, and the only error source is the estimation of the Hessian matrix using public data. For more complex models, a number of popular unlearning methods approximate full retraining by taking a Newton update. When we apply our attack to more complex models, we act as if full retraining results in the model that is derived from taking a newton step --- and the reason our reconstruction performance degrades is simply that this approximation becomes imperfect. However, if rather than full retraining, the machine unlearning method employed was one that simply took a newton step, our reconstruction would again be near perfect. We will elaborate on this point in the revision.
>
> Re: multi-sample deletion, our approach recover an approximation of the gradient sum for all the deleted samples. A determined adversary can poison an unlearning round by requesting deletion of n-1 samples known to the adversary, and so can recover the gradient of the single unknown point from this sum. Nevertheless we agree that multi-sample recovery is an important question that our work does not address, and think this is one of the most interesting questions for future work arising from our paper.
>
> On the displayed metrics: We show both the full distribution of cosine similarity as well as randomly selected reconstructions for visual inspection. While other metrics such as similarity on a well-chosen embedding function could be of interest, we chose to simplify the presentation and show the more challenging (pixel-wise, for images) cosine similarity comparison. We are open to suggestions and are happy to engage if you have particular additional metrics that you think would be informative.
>
> On defenses: Our work highlights the privacy risk of unlearning in standard models. As we discuss in our work,  training models using differential privacy (at small epsilon values) provably prevents reconstruction, even in unlearning scenarios because of differential privacy's composition property. There is an exciting opportunity for research suggested by our work: is there a way to give unlearning methods which have meaningful privacy guarantees even when the additional model training procedure did not?
>
> ----
> *[Response to Questions]*
>
> On the use of “Avg” as a baseline: This is a straightforward way of leveraging information about the data distribution in a manner that does not incorporate any information about the model parameters. The “MaxDiff” baseline is also included, and this is motivated by the assumption that the overall performance of the update on held-out data would be, in relative terms, much smaller than the peformance difference of the sample being deleted (since this sample transitions from being in the training distribution, to being outside it)
>
> On the robustness to model architectures: Our existing experiments already show results for a variety of simple models and loss functions on the same datasets (linear and logistic regression, ridge regression, ridge regression over random Fourier features, cross-entropy minimization over random Fourier features, as well as support vector machines over both raw features and random Fourier features).
>
> On the computational complexity of the attack: Our attack in its most general form relies on a Hessian-vector product per public sample as shown in Eq. 13. This can be efficiently implemented in all common deep learning frameworks with a computational complextiy of $O(nd^2)$ with $n$ being the number of public samples and $d$ the number of parameters in the model. It is also possible to replace the Hessian computation altogether by using the Fisher information matrix, bringing the computational complexity down to $O(nd)$.
>
> On countermeasures: The most straightforward countermeasure would be to train the original and updated models using differential privacy. This would provably prevent any reconstruction attack (as privacy composes across pairs of models if they are both trained privately). An exciting research direction suggested by our work is the study of unlearning methods in which the parameter update is itself differentially private with respect to the deleted samples --- and a fuller understanding of how this trades off with other unlearning desiderata.

---

> > ### Comment · Reviewer_VcL6 · 2024-08-13
> >
> > Thank you for your detailed response and clarifications. I have carefully reviewed your feedback and do not have any further questions at this time. I will maintain my current (positive) rating.

---

> > > ### Author Response · Authors · 2024-08-13
> > > **Thanks!**
> > >
> > > Thank you for engaging with our response and for your positive rating --- we appreciate it!

---

### Official Review · Reviewer_b41V · 2024-07-15

**Soundness:** 2
**Presentation:** 3
**Contribution:** 3
**Rating:** 5
**Confidence:** 3

**Summary:**

This work focuses on investigating privacy issues in machine unlearning. Specifically, assuming the availability of model parameters before and after unlearning, as well as the ability to sample data from the original data distribution, the proposed reconstruction attack aims to recover deleted samples. By analyzing the training objective of linear regression, the study found that the difference between the parameters before and after unlearning is proportional to the deleted sample. Based on this observation, this study proposes an algorithm to extract deleted samples accurately. The method extends to more complex models and arbitrary loss functions using Newton's method to approximate the parameter update process.

**Strengths:**

1. The topic of the study, concerning privacy risks in unlearning, is crucial. Since the data deleted in unlearning usually has high privacy sensitivity, recovering such data poses a significant threat.

2. The proposed algorithm is elegant and achieves near-perfect results in linear regression. It also has the potential to extend to more complex models.

**Weaknesses:**

1. The assumptions are too strong. The authors assume access to model parameters before and after unlearning and the ability to sample from the original data distribution. The authors need to clarify under what circumstances an attacker could have the assumptions mentioned in the paper, especially the sampling ability, since the deleted data is typically highly sensitive or inappropriate, making sampling difficult.

2. The goal of this paper is to explore the privacy risks in machine unlearning. In real-world scenarios, machine unlearning is achieved through existing unlearning methods. However, in this paper, unlearning is implemented by retraining after removing the samples to be unlearned. This creates a gap between the resulting model and the model obtained through actual unlearning methods. Therefore, it is questionable that the experimental results obtained from such a model can directly demonstrate the privacy risks associated with machine unlearning in real-world situations.

**Questions:**

1. In what scenarios is the threat model proposed in the paper realistic?
2. Is the proposed method effective when unlearning is performed using existing unlearning methods?

---

> ### Author Rebuttal · Authors · 2024-08-06
>
> *[Response to Weaknesses 1]*
>
> We take the position that security assurances should be based on minimal assumptions. Here, we view the assumption that an attacker who has API access to the model does -not- have access to model parameters to be dangerously strong. Consider for example a d dimensional linear model, as we study initially in our work. Just from query access, an attacker can recover the model parameters by querying the model on d linearly independent points and solving a system of linear equations. This requires no knowledge of the data distribution. So, there is no meaningful difference between white box and black box access in such scenarios.
>
> What about for more complex models? Even there the boundary between white-box and black-box access has been blurring. For example recent work [1] has shown that it is possible to reconstruct the embedding matrix from a production LLM model with only black-box access. And of course, open source models explicitly release parameters with each version of the model.
>
> With regards to access to the data distribution, we only assume access to samples from the same distribution as training data, not to private samples atually used in training. This is a standard assumption underlying even simpler attacks suck as membership inference. We agree with the reviewer that it may sometimes be difficult for an attacker to ascertain what this distribution is, but again, we take the view that when analysing attacks as a means to audit security vulnerabilities, we should be generous (within reason) about the assumed abilities of the attacker.
>
> [1] Carlini, Nicholas, et al. "Stealing part of a production language model." arXiv preprint arXiv:2403.06634 (2024).
>
> *[Response to Weaknesses 2]*
>
> An important aspect of existing machine unlearning approaches is that almost all of them aim to approximate full retraining with reduced computational cost. These approximations are proposed due to the intense compute required for full retraining, especially for large neural networks and LLMs.
>
> Our study is explicitly focused on exposing the risk present in even very simple models, such as linear regression, logistic regression, SVMs, and feature augmentation using random Fourier features. For such models, full retraining is feasible, and would be the expected solution to the data deletion problem, as computational approximations to retraining are not needed.
>
> Full retraining for a single data deletion in linear regression is equivalent to updating the model parameters using Newton's update as we discussed in Sec. 3.2. By leveraging this concept, our attack achieves almost perfect reconstruction, and the only error source is the estimation of the Hessian matrix using public data. For more complex models, a number of popular unlearning methods approximate full retraining by taking a Newton update. When we apply our attack to more complex models, we act as if full retraining results in the model that is derived from taking a newton step --- and the reason our reconstruction performance degrades is simply that this approximation becomes imperfect. However, if rather than full retraining, the machine unlearning method employed was one that simply took a newton step, our reconstruction would again be near perfect. We will elaborate on this point in the revision.

---

> > ### Comment · Reviewer_b41V · 2024-08-11
> >
> > (1) I partially agree with the author's viewpoint that when studying security risks, assumptions about the attacker's capabilities should be generous. However, if the assumed attacker's capabilities are too strong, the significance of the proposed method may decrease accordingly.
> >
> > (2) I still think that perfect unlearning is impossible in real-world scenarios, and what is usually achieved is an approximation. If this approximation leads to a decrease in the effectiveness of the proposed method, it would undoubtedly undermine its value.
> >
> > Based on the above reason, I raised my score to 5.

---

> > > ### Author Response · Authors · 2024-08-11
> > > **Thanks**
> > >
> > > Thanks --- we appreciate your engagement.

---

### Decision · Program_Chairs · 2024-09-25

**Decision:**

Accept (poster)

**Comment:**

The paper studies the leakage introduced by machine unlearning in simple settings, and shows that entire data points can be reconstructed.
While I agree with reviewers that the setting is not entirely realistic, I think this type of "bottom-up" work is valuable to gain a better understanding of the leakage of machine unlearning.